# Effects of Four Weeks of Plyometric Training Performed in Different Training Surfaces on Physical Performances in School Children: Age and Sex Comparisons

**DOI:** 10.3390/children9121914

**Published:** 2022-12-07

**Authors:** Hamza Marzouki, Ibrahim Ouergui, Rached Dridi, Okba Selmi, Rania Mbarki, Nour Mjadri, Mabliny Thuany, Marilia S. Andrade, Ezdine Bouhlel, Katja Weiss, Beat Knechtle

**Affiliations:** 1High Institute of Sports and Physical Education of Kef, University of Jendouba, Kef 7100, Tunisia; 2CIFI2D, Faculty of Sports, University of Porto, 4200-450 Porto, Portugal; 3Departamento de Fisiologia, Universidade Federal de São Paulo (UNIFESP), São Paulo 04021-001, SP, Brazil; 4Laboratory of Cardio-Circulatory, Respiratory, Metabolic and Hormonal Adaptations to Muscular Exercise, Faculty of Medicine Ibn El Jazzar, University of Sousse, Sousse 4000, Tunisia; 5Institute of Primary Care, University of Zurich, 8006 Zurich, Switzerland; 6Medbase St. Gallen Am Vadianplatz, 9000 St. Gallen, Switzerland

**Keywords:** strength training, explosive performance, high-intensity performance, sex, age

## Abstract

Short- to middle-term plyometric training has been shown to be an effective method to promote youth fitness and health. However, there is no knowledge of previous studies that investigated the sex and age effects on physical fitness following different PT surfaces (i.e., firm vs. sand) in schoolchildren. This study examined the effects of age and sex on explosive and high-intensity responses following plyometric training (4 weeks, twice/week) performed on firm vs. sand surfaces in untrained schoolchildren. Ninety girls and ninety boys (under 8: age = 7.1 ± 0.5 and 7.1 ± 0.4 years; under 10: age = 9.0 ± 0.4 and 9.0 ± 0.5 years; under 12: age = 11.0 ± 0.5 and 11.0 ± 0.5 years, respectively) participated in a randomized and parallel training design with pre-to-post testing. Participants were allocated (i.e., 30 boys and 30 girls for each group) into either two experimental groups (firm group: performing plyometrics on a clay surface and sand group: performing plyometrics on a dry surface of 20 cm deep sand) or a control group (CG, habitual physical education classes) within their corresponding age groups. Children were tested for sprint, jumping and change of direction speed performances before and after 4 weeks of plyometric training. Both experimental groups induced more significant improvements in all assessed variables than CG (*p* < 0.0001; effect size > 0.80), whereas both surfaces induced similar improvements (*p* > 0.05). Older boys achieved better performances than their younger counterparts (*p* < 0.05) and older girls (*p* < 0.0001), respectively. This finding showed that age and sex could affect explosive and high-intensity performances during childhood after a short-term plyometric training. In contrast, the training-induced fitness changes were not influenced by the type of surface.

## 1. Introduction

It is commonly known that children can develop motor skills by engaging spontaneously in regular physical activities by performing various drills such as running and jumping [1]. Jumping activity is usually associated with plyometric training (PT), where fast and forceful multi-joint motions such as leaping, hopping, and skipping are involved [2]. These motions are also conditioned using the stretch-shortening cycles (SSC), where muscles are stretched quickly and then rapidly contracted [3]. The SSC is of great importance in plyometric training based on its participation in improving the muscle-tendon unit’s capacity and generating the maximal force in the shortest time [4]. 

Previously, PT was considered unsafe for children due to the musculoskeletal soreness, discomfort, and articular overloading caused by the ground reaction forces during jumps [5]. However, recent studies have reported that by respecting the principle of specificity and using good technique and safety measures, PT may be used effectively to improve children’s fitness [2,6]. Likewise, prior studies have reported that short- to middle-term PT has been shown to be an effective training method to enhance linear sprinting, lower body muscular power, jump, and change of direction speed (CODS) performances in children aged 7 to 12 years [2,7,8]. 

To prescribe effective PT programs, several variables should be respected, such as the safety, type, number and intensity of jump drills, the overload principle, and the training surface [9,10,11,12]. Considering this, previous studies have indicated that the magnitude of the PT effect on athletic performances is influenced by the amount of energy returned to the athlete from the training surfaces, which mainly depends on the training surface stiffness [9,13,14,15,16,17]. Therefore, the surface-type (e.g., hard, grass, sand) is an essential factor that can condition the effectiveness of PT through the stretch-reflex mechanism [18]. This research was performed only in youth [15] or adolescent male athletes [13,14]. Although previous training protocols were performed with children [7], to the best of our knowledge, no previous reports have examined the sex and age effects following different PT surfaces in children. Since PT generates dynamic movements and greater strength on muscles and bones, it could be an effective strategy to further increase the child’s motor ability to run, hop, throw, and kick, which are extensively used in recreational and playing activities such as dancing, swimming, gym workouts [19].

The purpose of this investigation was to examine the impact of PT (4 weeks, twice/week) performed on firm vs. sand surfaces on explosive (i.e., sprinting, jumping) and high-intensity (i.e., CODS) performances in male and female schoolchildren of different age groups [under 8 (U8), under 10 (U10) and under 12 (U12) years old]. We hypothesized that surface-type PT could positively influence the explosive and high-intensity performances more than the habitual routines of physical education lessons (Control group: CG), with different improvements on both surfaces. We also hypothesized that older boys and girls would perform better than their younger counterparts.

## 2. Materials and Methods

### 2.1. Participants

A sample size estimation was conducted using ANOVA test (fixed effects, special, main effects, and interactions), with sex (boys and girls) × age [U8, U10 and U12] × condition [firm (FG), or sand (SG), or control]. The effect sizes used to generate the sample size estimation were attained from previous investigations exploring variables such as the proposed research [14]. The calculation resulted that a total sample size of 193 participants would be sufficient to detect differences (effect size f = 0.25, α = 0.05, 1 − ß = 0.80) with an actual power of 80.20%. Thus, one hundred and six girls and ninety-nine boys volunteered to participate in the study. Fliers and advertisements about the study were used to recruit the participants with their parent’s approval. Random allocation was maintained with simple randomization stratified by sex and age, which resulted in the following assignments: two experimental groups [firm surface group (FG): performing PT on a clay surface, 39 girls and 35 boys; sand surface group (SG): performing PT on a dry surface of 20-cm deep sand, 35 girls and 33 boys)] and a control group (32 girls and 31 boys). To be included in the final analysis, the participants should complete at least 90% of the total training sessions, which resulted in excluding the data of 25 participants (i.e., 16 girls and 9 boys) from the final analyses. Consequently, 90 girls and 90 boys were included in the final analyses from three different age groups, including 60 from each age group (U8, U10 and U12). There were no significant differences in anthropometric or performance variables between groups with the same age at the beginning of the protocol for either gender. Characteristics of participants from different groups are included in Table 1. The exclusion criteria from the study were: disability, physical effort contraindicated by a doctor, comorbidities affecting the body function (e.g., asthma, cardiorespiratory failure, cardiovascular diseases) and having participated in any physical training program during the period of the experimental approach. All participants were screened and were safe from injuries prior to preliminary testing. Before participating in the study, parental informed consent was obtained as participants were younger than 18 years. The study was conducted in accordance with the Declaration of Helsinki [20] and approved by the Research Ethics Committee of the High Institute of Sport and Physical Education of Kef (UR13JS01), University of Jendouba, Tunisia (protocol code 013/2020; date of approval: 7 May 2020).

### 2.2. Procedures

A randomized, parallel, and controlled pre-to-post measurement trial was adopted in this study. Two hundred and five children aged 6 to under 12 years from a similar socio-economic status, living in the same city and studying in the same public primary school, volunteered for this investigation. All schoolchildren regularly participated in their normal physical education classes twice per week (1 h per session). Participants were not involved in any extra-school activities, including strength and conditioning training programs, 6 months prior to participating in the investigation. The study was conducted between October and November of the scholar year 2020–2021, lasting six weeks and consisting of one week of pre-testing (T1), four weeks of PT, and one week of post-testing (T2). Experimental and control groups were tested for the sprint, jumping, and CODS performances.

One week before starting the experimentation, all children were familiarized with the experimental procedures to reduce the impact of the learning effect on testing outcomes. The results of the tests during the familiarization session and T1 were also used for assessing the test-retest reliability of the measures. Participants were requested not to exert any vigorous effort on test day or the day preceding the assessments or training. All tests were performed at the same time of the day under similar environmental conditions (indoors in the school gymnasium on the floor, temperature: 20–26°, humidity ≈ 54 ± 3%). During all testing and training sessions, similar verbal encouragements were provided to all subjects by the investigators. Before starting the tests, participants exercised a 15-min standard warm-up, including 10 min of jogging and light stretching, followed by 5min of intense drills (short-sprint, skipping, leg and arm swings). The linear sprint, jumping and CODS tasks involved two valid maximal trials interspersed with 2-min passive recovery, and the best trial was used for analysis. Five min of rest was allowed between tests. The PT consisted of twice-weekly exercise sessions on non-consecutive days for four weeks, exercised after the physical education session warm-up. Immediately after completing each PT session, the experimental groups were requested to join CG in order to complete the usual physical education class. Across the training study, CG performed only their normal physical education sessions (i.e., relay games, gymnastics, ballgames).

### 2.3. Measurements

#### 2.3.1. Anthropometric Measurements

At the start of the first testing session (T1), anthropometric parameters of body height (cm) and body mass (kg) were assessed twice for each participant, and the mean of each measure set was calculated. Body weight was measured to the nearest 0.1 kg using a digital scale (OHAUS, Florhman Park, NJ, USA), body height was measured to the nearest 0.1 cm, and body mass index [BMI] was determined [kg·m^−2^].

#### 2.3.2. Sprint Testing

The sprinting performance was evaluated using the 10 m standing-start all-out run (S10). In this test, velocity was measured by the mean of photocell gates (Globus, Microgate, Bolsano, Italy) placed 1.0 m above the ground at the starting point and on the 10 m lines, with a marker for the front foot placed 5 cm behind the start position. The participants were instructed to run at their maximum speed until the stop line.

#### 2.3.3. Vertical Jumping

Participants performed the countermovement jump (CMJ) test without arm swing, beginning from an upright standing position and performing a very fast preliminary downward eccentric action followed immediately by a powerful upward motion to jump as high as possible [21]. The jumping height (cm) was assessed with an infrared jump system (Optojump; Microgate, Bolzano, Italy).

#### 2.3.4. Change of Direction Speed Testing

The 4 × 10 m shuttle run test (SHT) was used to evaluate the children’s CODS [22]. The participants’ velocity was measured using the photocell gates, placed 1.0 m above ground at the start/finishing line. Two parallel lines were drawn on the floor 10 m apart for the running and turning (shuttle) test at maximum speed (4 × 10 m). Participants were instructed to run and turn as fast as possible, crossing each line with both feet every time, covering a distance of 40 m.

### 2.4. Plyometric Training Programs

The training intervention duration was selected based on previous research, which reported positive adaptation of SSC measures after four weeks of plyometric training [23]. To minimize the risk of possible injuries’ occurrence, training sessions’ loads progressed from low to moderate-high intensity drills (i.e., by the increase in jumps and hops ‘number as well as the level of exercise complexity), imposing, therefore, gradually greater eccentric stress on the musculotendon unit, as previously reported [24]. Specifically, each training session was composed of 4–6 different exercises with 2–4 sets of 4–10 repetitions (Table 2). Plyometric drills lasted approximately 5–10 s, and at least 90 s of passive recovery were allowed after each set. Children were instructed to perform maximal efforts with brief ground contact times. An exercise was repeated when it was not performed correctly. Before each training session, a 10-min standardized warm-up was performed, consisting of low-intensity aerobic activity and sets of mobility exercises that provided appropriate activation of the lower limb muscles [23]. No injuries or damage occurred during the training period.

### 2.5. Statistical Analysis

Descriptive outcomes were presented as means ± standard deviation (SD). The intraclass correlation coefficient (ICC) and standard error of measurement (SEM) were used to test the relative and absolute reliability of each outcome measure. Pre-to-post change (Δ%) was calculated for each physical variable and was used for the subsequent analyses. Shapiro-Wilk (normality of the distribution) and Levene’s (homogeneity of variance) tests were calculated for all experimental data before inferential testing. A 3-way univariate analysis of variance (ANOVA) (sex [boys and girls] × age [U8, U10 and U12] × condition [FG, or SG, or CG]) was conducted to evaluate age and sex effects on variables’ pre-post change in response to surface-type plyometric training. If significant main effects or interactions were found, a Bonferroni post-hoc analysis was conducted. Effect sizes (ES) were determined by converting the partial eta squared to Cohen’s d [25] to determine the magnitude of differences. The magnitude of effect size was classified as trivial (<0.20), small (0.20–0.49), medium (0.50–0.79), and large (0.80 and greater) [25]. The analyses were performed using SPSS version 20 for Windows (IBM Corp, Armonk, NY, USA) and the significance level was set at 0.05.

## 3. Results

Normality of the data and the homogeneity of variance were confirmed. The ICC and SEM values for the test-retest trial were 0.929 and 2.398% for S10, 0.976 and 8.055% for CMJ, and 0.986 and 2.129% for SHT, respectively, indicating a good to excellent reliability. After the training period, all groups improved their athletic performances (all *p* < 0.05) (Table 3 and Table 4). Both FG and SG showed more significant pre-post changes in S10, CMJ and SHT performances than CG (all *p* < 0.0001 and ES > 0.80) (Table 3 and Table 4). However, physical performances had similar improvements on both surfaces (all *p* > 0.05).

### 3.1. Sprint Testing

For S10 performance, there was no interaction between age × condition and sex × age × condition (Table 3). However, a significant interaction effect between sex and condition was found (Table 3). There was a significant difference between sex groups with the U12-year-old boys resulted in better speed performances than the U12-year-old girls on firm (*p* < 0.0001; ES = 2.319) and sand (*p* < 0.0001; ES = 2.360) surfaces. Moreover, when boys were pooled across age, the pre-to-post change was significantly higher in U12 than U10 (on firm: *p* < 0.0001 and ES = 1.662; on sand: *p* < 0.0001 and ES = 1.807) and U8 (on firm: *p* < 0.0001 and ES = 2.214; on sand: *p* < 0.0001 and ES = 2.123).

### 3.2. Vertical Jumping

An interaction between sex and condition was significant for the pre-to-post changes in the CMJ test. However, no interaction effect was found between sex, age and condition (Table 3). There was a significant difference between sex groups, with the U12-year-old boys resulted in higher CMJ performances than U12-year-old girls on firm (*p* < 0.0001; ES = 4.358) and sand (*p* < 0.0001; ES = 3.155) surfaces. Moreover, a significant age × condition interaction effect was found (Table 3), with the U12-year-olds showed better performances than the younger age groups either on firm (U12 vs. U10: *p* < 0.0001 and ES = 2.444; U12 vs. U8: *p* < 0.0001 and ES = 2.241) or sand (U12 vs. U10: *p* < 0.0001 and ES = 3.323; U12 vs. U8: *p* < 0.0001 and ES = 1.585) surfaces within boys.

### 3.3. Change of Direction Speed Testing

For SHT performance, a significant sex × condition and age × condition interaction effect was found (Table 4). On the one hand, when participants were pooled across sex, pre-to-post change was significantly greater for U12-year-old boys than U12-year-old girls on firm (*p* < 0.0001, ES = 4.505) and sand (*p* < 0.0001; ES = 5.753). On the other hand, when boys were pooled across age, pre-to-post change was significantly higher in U12 than U10 (on firm: *p* < 0.0001 and ES = 4.624; on sand: *p* < 0.0001 and ES = 4.878) and U8 (on firm: *p* < 0.0001 and ES = 6.039; on sand: *p* < 0.0001 and ES = 6.242). Moreover, U10-year-old boys showed better improvements than U8-year-old boys (on firm: *p* = 0.001; ES = 1.671; on sand: *p* < 0.0001; ES = 2.284). In addition, when girls were pooled across age, pre-to-post change was significantly higher in the U12 and U10-year-olds than U8-year-olds (on firm: all *p* < 0.0001, ES = 2.334 and ES = 2.237, respectively; on sand: all *p* < 0.0001, ES = 3.015 and ES = 2.880, respectively).

## 4. Discussion

The present study revealed that experimental groups (i.e., FG and SG) induced more significant enhancements in the S10, CMJ and SHT tests than the CG groups. However, both surfaces induced similar improvements, which disagrees with the second part of the first hypothesis. The results of this study also showed that older boys achieved better performances than their younger counterparts and older girls, respectively. Except for SHT, explosive performances were not affected by age in girls, which partially confirms our second hypothesis.

Our results agree with previous studies showing significant improvements in sprinting, jumping ability and CODS among children in response to plyometric training [2,7,8,23,26,27]. It has been reported that children may successfully benefit from training interventions focusing intensively on developing physical skills [19] through the refinement of the cortical network, which positively impacts motor abilities, speed, muscular strength, movements accuracy, and motor coordination during normal neuromotor development [28]. In the present study, the improvements induced by PT programs compared to the non-plyometric training may be explained by different training-related mechanisms such as the changes in the musculotendinous stiffness, the maximal isometric voluntary force increase, type I to type II muscle-fiber transition, the increase in the muscle contractility magnitude, muscle size increase, altered fascicle angle, enhanced motor unit recruitment and discharge rate, greater inter-muscular coordination, higher stretch-reflex excitability, an enhanced neural drive of the agonist muscle, better utilization of the stretch-shortening cycle, and rate of torque development in the quadriceps of children [29,30]. Thus, it seems that training programs including movements that are biomechanically and metabolically specific to the subsequent performance test (i.e., CMJ) may be more likely to induce improvements in selected performance measurements [6,31]. In addition to the positive effects of our PT protocols on athletic performance, the additional gains were achieved without any musculoskeletal injury. This supports the recommendations that stipulate the need for children to participate in various strength and conditioning programs, driven by technical competence and appropriate to age, in order to facilitate athletic development [32]. Our findings suggest that adding PT (performed twice a week with low to moderate-high intensity drills) to regular physical education sessions can be a safe strategy and positive stimulus to improve explosive and high-intensity performances in healthy schoolchildren.

Similar effects were found in the present study when physical fitness ‘improvements across PT programs using sand and firm surfaces were compared. The present study reported similar findings to those previously shown in different youth populations [9,13,14,15,17]. More specifically, four to eight weeks of PT performed on sand produced similar sprint, jump and CODS performance improvements compared to those observed after training on firm surfaces [9,13,14,15,17]. In that regard, it has been outlined that accomplishing explosive tasks (i.e., sprinting, jumping) on firm surfaces can improve the ability of muscles to utilize stored elastic energy during the eccentric phases generating fast and powerful concentric actions [17,33,34,35]. Thus, the positive adaptations induced by firm surfaces may be more related to improved efficiency in storing and reusing elastic energy during explosive actions [9]. Conversely, it has been shown that sand training may induce a considerable amount of elastic energy dissipation, increasing the energy cost and the level of muscle activation [36,37], and might represent an alternative to increase overload during training [9]. Given that both modalities are easy to implement and effectively enhance athletic performances, it is conceivable to suggest that a range of low to moderate volume sessions of these two distinct and possibly complementary mechanisms can be scheduled prior specific physical activities (e.g., as warm-up drills) [9].

Interestingly, sprint, CMJ and CODS performance changes were age group dependent, with U12-year-olds resulting in more significant improvements in all variables than their younger counterparts. This would suggest that age and maturation status have an effect on the natural development of SSC performance, irrespective of training intervention, a finding that has been previously reported by other studies [23,24]. Our findings revealed that U12-year-old boys produced greater explosive and high-intensity values than both U10- and 8-year-old boys. Moreover, U10-year-old boys produced better results for all variables than 8-year-old boys. These findings reflect the influence of age on training adaptation, which could be considered a normal and expected adaptation. Furthermore, it was stated that the gain in both body mass and height with age could be associated with improved running speed, vertical jump, and CODS [7]. Similarly, since muscular strength increases progressively with both body weight and stature, this increase can be extended to the other variables [38]. Testosterone in men promotes the development of sexual function and somatic characteristics [39]. This hormone may stimulate the anabolic effect of the insulin-like growth factor 1 on muscle cells as well as facilitate neuromuscular transmission, thus improving cognitive effects [39].

Regarding the effects of age on CODS improvements induced by PT, older children (i.e., U12 and U10) showed a greater CODS performance compared to younger children (i.e., U8) in either boys or girls. Regarding training at different ages, it seems that maturation plays a crucial role in CODS performance gains in response to PT [40]. It was reported that children between 10 and 12 years showed accelerated SSC development, a phenomenon that continued near peak height velocity [24,41]. It might be that older children express greater plasticity after PT in muscle size, the transition from type I to type II muscle fibers, muscle contractile ability, fascicle angle, motor unit recruitment, intermuscular coordination, stretch-reflex excitability, utilization of the SSC properties, and neural drive to agonist muscles [29]. Additionally, the maturation-related development of the central nervous system may lead to further enhancements in CODS after PT for older children compared with their less mature peers [40,42], and this affects their responses to learning and training stimuli [43]. During growth, children develop their motor skills (e.g., coordination, agility, balance) as their nervous systems develop, a developmental process that continues well beyond puberty (e.g., myelination of nerve fibers) [44]. The completion of myelination accelerates the conduction of impulses along the neurons, thus allowing for the production of skillful movements [44]. Motor performance could be improved following the practice of an activity or a skill whose complete development depends on myelination [44]. In this respect, the incomplete myelination could influence CODS development. However, future studies are needed to clarify which of these mechanisms may help to explain these results more effectively. Nevertheless, no significant differences were found in sprinting and jumping ability gains when girls were compared by age groups. The reasons that can explain why these two variables were not affected by age remain unclear and it would be speculative to provide an explanation.

Comparisons between the two sexes revealed that U12-year-old boys had better training-induced explosive and high-intensity adaptations than girls from the same age group on either FG or SG. These data contrast the results of previous studies conducted with prepubescent children (aged 10.3 to 10.9 years), which reported no significant sex differences following PT [45,46,47]. The results of the present study can be explained by the fact that possibly U12-year-old girls were entering puberty, which is characterized by increased estrogen secretion, which induces adipose tissue increase [48]. The fat mass represents an inert noncontributory load and thus an increased metabolic cost for children, making them less efficient in cardiorespiratory response and their performance during tasks where the body must be projected [49]. Likewise, the noncontributory mass could lead to biomechanical movement inefficiency and could be detrimental to motor proficiency [50]. In other words, these simultaneous processes can reduce the female’s relative strength and, consequently, her capacity to jump higher and optimize adaptations to PT. The fact that sex could significantly affect plyometric training-induced explosive and high-intensity adaptations at the prepubertal period should be considered to optimize well-rounded training programs in schools.

The current novel findings are not without limitations that should be considered to interpret the results with caution. First, the investigation lasted only four weeks, whereas longer periods of PT may be required to achieve greater physical fitness performance differences between SG and FG. Second, due to the limited testing procedure (e.g., no electrophysiological measures, no leg stiffness and muscle damage assessments), it was not possible to clarify the mechanisms underlying the observed effects. Thus, future studies should be conducted to incorporate physiological and biomechanical analyses to better understand the potential underlying mechanisms of the changes observed in the physical fitness variables analyzed in boys and girls during childhood. In particular, developmental characteristics should be considered when planning children’s physical exercise protocols to respect neuromotor plasticity [7]. Third, reference should be made to the different types of sand (grammage, washed sand or not, wet, or not, etc.). This is undoubtedly a limiting factor and can cause bias. It should therefore be reflected as an important aspect. Finally, different training program designs or different methods of organizing training workouts (e.g., plyometrics performed in water, plyometrics combined to sprint training program) may have led to different training-induced outcomes.

## 5. Conclusions

As hypothesized, FG and SG showed higher enhancements in S10, CMJ and CODS tests than CG. This finding about motor performance in children is of particular interest to practitioners and educators because educational term times are typically short, and children are likely to respond more favorably to regular changes in a periodized training program [51]. Additionally, our results revealed that training on the sand surface produced similar explosive and high-intensity performance improvements to those observed after training on a firm surface. Practitioners should be aware that sand training programs may be a suitable and alternative strategy to be incorporated into their weekly routines in conjunction with more traditional training practices, such as plyometric exercises on harder surfaces [9]. More importantly, the results of this study showed that age and sex could affect pre-to-post change in explosive and high-intensity performances during childhood after a short-term PT. This finding is of particular interest and suggests that practitioners and physical education teachers should consider the principle of individualization to enhance the efficiency of their educational programs.

## Figures and Tables

**Table 1 children-09-01914-t001:** Descriptive data of anthropometric and demographic parameters for the surface-type plyometric training and control groups at pre-test (*n* = 180).

Age Group	Sex	Condition (*n*)	Age (Years)	Height (cm)	Weight (kg)	BMI (kg·m^−2^)
U8	Boys	FG (10)	7.2 ± 0.4	125.1 ± 0.0	22.5 ± 1.5	14.4 ± 0.5
		SG (10)	7.0 ± 0.5	123.9 ± 0.1	21.9 ± 3.8	14.2 ± 1.6
		CG (10)	7.1 ± 0.5	124.6 ± 0.0	21.7 ± 2.9	13.9 ± 1.5
	Girls	FG (10)	7.1 ± 0.6	125.0 ± 0.0	22.8 ± 2.1	14.8 ± 1.1
		SG (10)	7.1 ± 0.5	124.8 ± 0.0	22.9 ± 4.3	14.6 ± 1.8
		CG (10)	7.2 ± 0.5	124.0 ± 0.0	22.0 ± 2.6	14.3 ± 1.4
U10	Boys	FG (10)	9.1 ± 0.4	135.6 ± 0.1	30.1 ± 4.4	16.3 ± 1.3
		SG (10)	8.9 ± 0.6	134.0 ± 0.0	28.9 ± 3.9	16.1 ± 2.1
		CG (10)	8.9 ± 0.5	134.8 ± 0.0	30.2 ± 3.8	16.6 ± 1.7
	Girls	FG (10)	9.0 ± 0.4	133.6 ± 0.1	30.7 ± 7.3	17.0 ± 3.0
		SG (10)	9.0 ± 0.6	135.3 ± 0.1	31.2 ± 6.8	16.9 ± 2.3
		CG (10)	9.1 ± 0.5	135.5 ± 0.1	31.5 ± 5.9	17.2 ± 3.6
U12	Boys	FG (10)	11.1 ± 0.5	150.6 ± 0.1	42.0 ± 10.3	18.4 ± 3.6
		SG (10)	11.0 ± 0.5	150.3 ± 0.1	41.9 ± 4.4	18.5 ± 0.9
		CG (10)	11.0 ± 0.6	148.6 ± 0.1	41.1 ± 8.3	18.4 ± 1.9
	Girls	FG (10)	11.1 ± 0.6	153.0 ± 0.1	45.5 ± 7.9	19.5 ± 3.4
		SG (10)	11.0 ± 0.4	150.6 ± 0.1	43.5 ± 4.8	19.1 ± 1.1
		CG (10)	11.1 ± 0.6	151.5 ± 0.1	46.1 ± 11.6	19.8 ± 2.9

Values are given as mean ± SD; U8: Under 8; U10: Under 10; U12: Under 12; FG: firm group; SG: sand group; CG: control group; BMI: body mass index.

**Table 2 children-09-01914-t002:** Description of the plyometric training program.

	Week 1		Week 2		Week 3		Week 4	
Type of Jumps	Session 1	Session 2	Session 1	Session 2	Session 1	Session 2	Session 1	Session 2
Pogo jump	2 × 6	2 × 6	2 × 8	2 × 10	2 × 10	4 × 8	4 × 8	4 × 10
Lateral jump	2 × 6	4 × 6	2 × 8					
Hopscotch	3 × 4							
Bilateral power hops	4 × 4	4 × 4	4 × 4					
Ankle hops	2 × 6	3 × 5	3 × 5	3 × 5				
Power skipping			2 × 6	2 × 8	3 × 8			
Unilateral pogo jump				2 × 8	2 × 10	2 × 8	2 × 8	2 × 10
Max rebound hops				3 × 5	3 × 5	3 × 5	4 × 5	
Drop jump					2 × 5	2 × 5	2 × 5	2 × 6
Hurdle power hops						2 × 6	2 × 5	2 × 5
Double tuck jumps						2 × 5	2 × 6	2 × 6
Alternating jump lunges								2 × 5
Total foot contacts	64	67	75	82	89	95	100	104

Number of Sets × Number of repetitions; 90-s of passive recovery between sets.

**Table 3 children-09-01914-t003:** Descriptive data for sprint (s) and jump (cm) performances in the surface-type plyometric training and control groups at pre-test and post-test (*n* = 180).

Variables	Age Group	Sex	Condition (*n*)	Pre-Test	Post-Test	Δ (%)	ANOVA Analysis	
							*F*-Value	ES (*p*-Value)
S10	U8	Boys	FG (10)	2.92 ± 0.09	2.78 ± 0.08	−4.5 ± 0.5 ¶	**- interaction** age × sex × condition: F_(4, 162)_ = 2.008 age × condition: F_(4, 162)_ = 1.019 sex × condition: F_(2, 162)_ = 3.814 age × sex: F_(2, 162)_ = 18.481 **- main effect** Condition: F_(2, 162)_ = 269.551 Age: F_(2, 162)_ = 15.282 Sex: F_(1, 162)_ = 34.969	
			SG (10)	2.91 ± 0.26	2.79 ± 0.26	−4.2 ± 0.9 ¶		
			CG (10)	2.91 ± 0.41	2.88 ± 0.40	−1.0 ± 0.3		0.155 (0.096)
		Girls	FG (10)	2.92 ± 0.14	2.80 ± 0.12	−4.2 ± 0.5 ¶		
			SG (10)	2.92 ± 0.18	2.79 ± 0.17	−4.4 ± 1.1 ¶		0.021 (0.399)
			CG (10)	2.91 ± 0.23	2.88 ± 0.23	−1.2 ± 0.5		
	U10	Boys	FG (10)	2.74 ± 0.19	2.62 ± 0.18	−4.4 ± 0.4 ¶		0.185 (0.024)
			SG (10)	2.72 ± 0.20	2.59 ± 0.19	−4.8 ± 1.0 ¶		
			CG (10)	2.72 ± 0.18	2.68 ± 0.18	−1.4 ± 0.4		0.460 (<0.0001)
		Girls	FG (10)	2.83 ± 0.24	2.71 ± 0.23	−4.1 ± 1.2 ¶		
			SG (10)	2.84 ± 0.16	2.72 ± 0.15	−4.3 ± 0.9 ¶		
			CG (10)	2.84 ± 0.06	2.81 ± 0.06	−1.1 ± 0.4		1.804 (<0.0001)
	U12	Boys	FG (10)	2.51 ± 0.19	2.40 ± 0.18	−4.5 ± 0.8 ¶†‡§		
			SG (10)	2.53 ± 0.24	2.42 ± 0.21	−4.4 ± 1.2 ¶†‡§		0.416 (0.011)
			CG (10)	2.50 ± 0.19	2.46 ± 0.19	−1.6 ± 0.5		
		Girls	FG (10)	2.54 ± 0.18	2.47 ± 0.19	−2.9 ± 0.8 ¶		0.455 (<0.0001)
			SG (10)	2.55 ± 0.13	2.48 ± 0.12	−2.9 ± 0.4 ¶		
			CG (10)	2.55 ± 0.16	2.52 ± 0.16	−1.2 ± 1.1		
CMJ	U8	Boys	FG (10)	13.1 ± 5.4	14.3 ± 6.1	9.7 ± 7.3 ¶	**- interaction** age × sex × condition: F_(4, 162)_ = 1.369 age × condition: F_(4, 162)_ = 5.752 sex × condition: F_(2, 162)_ = 3.987 age × sex: F_(2, 162)_ = 9.526 **- main effect** Condition: F_(2, 162)_ = 234.654 Age: F_(2, 162)_ = 4.644 Sex: F_(1, 162)_ = 11.283	
			SG (10)	12.5 ± 5.3	13.7 ± 5.8	9.6 ± 7.3 ¶		
			CG (10)	13.2 ± 2.8	13.6 ± 2.7	3.5 ± 2.7		0.097 (0.238)
		Girls	FG (10)	11.6 ± 4.7	12.7 ± 5.3	9.1 ± 6.7 ¶		
			SG (10)	11.8 ± 4.9	12.9 ± 5.5	9.1 ± 7.3 ¶		0.037 (<0.0001)
			CG (10)	12.3 ± 2.7	12.7 ± 2.7	3.5 ± 2.7		
	U10	Boys	FG (10)	15.4 ± 2.2	16.9 ± 2.2	10.0 ± 2.1 ¶		0.190 (0.020)
			SG (10)	15.1 ± 1.5	16.6 ± 1.7	9.8 ± 0.7 ¶		
			CG (10)	16.2 ± 0.8	16.6 ± 0.9	3.0 ± 0.5		0.321 (<0.0001)
		Girls	FG (10)	13.7 ± 2.0	15.0 ± 1.8	9.8 ± 3.4 ¶		
			SG (10)	13.9 ± 2.7	15.2 ± 2.4	9.8 ± 2.0 ¶		
			CG (10)	13.9 ± 1.6	14.4 ± 1.6	3.8 ± 1.4		1.681 (<0.0001)
	U12	Boys	FG (10)	18.2 ± 2.2	20.7 ± 2.4	13.8 ± 0.8 ¶†‡§		
			SG (10)	18.1 ± 2.0	20.7 ± 2.3	13.9 ± 1.6 ¶†‡§		0.210 (0.011)
			CG (10)	17.3 ± 2.5	17.7 ± 2.5	2.4 ± 1.4		
		Girls	FG (10)	17.1 ± 1.5	18.7 ± 1.6	9.5 ± 1.2 ¶		0.250 (<0.0001)
			SG (10)	16.5 ± 1.1	18.1 ± 1.3	9.6 ± 1.1 ¶		
			CG (10)	17.2 ± 1.1	17.5 ± 1.0	2.1 ± 0.9		

Values are given as mean ± SD; U8: Under 8; U10: Under 10; U12: Under 12; FG: firm group; SG: sand group; CG: control group; Δ (%): pre-post change percentage; ES: effect size; S10: 10-m sprint; CMJ: countermovement jump. ¶ Significantly different from CG. † A significant difference when comparing boys to girls of FG and SG at U12, respectively. ‡ A significant difference when comparing U12 boys to U8 boys of FG and SG, respectively. § A significant difference when comparing U12 boys to U10 boys of FG and SG, respectively. The statistical significance level was set at *p* ≤ 0.05.

**Table 4 children-09-01914-t004:** Descriptive data for change of direction speed performance (s) in the surface-type plyometric training and control groups at pre-test and post-test (*n* = 180).

Variables	Age Group	Sex	Condition (*n*)	Pre-Test	Post-Test	Δ (%)	Anova Analysis	
							*F*-Value	ES (*p*-Value)
SHT	U8	Boys	FG (10)	15.9 ± 0.6	15.0 ± 0.6	−5.3 ± 0.6 ¶	**- interaction** age × sex × condition: F_(4, 162)_ = 18.291 age × condition: F_(4, 162)_ = 43.111 sex × condition: F_(2, 162)_ = 18.185 age × sex: F_(2, 162)_ = 59.691 **- main effect** Condition: F_(2, 162)_ = 930.927 Age: F_(2, 162)_ = 148.827 Sex: F_(1, 162)_ = 59.691	
			SG (10)	15.8 ± 0.6	15.0 ± 0.6	−5.1 ± 0.6 ¶		
			CG (10)	16.0 ± 0.8	15.5 ± 0.7	−2.9 ± 0.4		0.644 (<0.0001)
		Girls	FG (10)	16.7 ± 0.6	15.9 ± 0.5	−5.0 ± 0.3 ¶		
			SG (10)	16.7 ± 0.8	15.9 ± 0.8	−4.8 ± 0.3 ¶		1.004 (<0.0001)
			CG (10)	16.4 ± 1.1	16.0 ± 1.1	−2.6 ± 0.4		
	U10	Boys	FG (10)	14.9 ± 0.9	14.0 ± 0.8	−6.2 ± 0.6 ¶¥		0.456 (<0.0001)
			SG (10)	14.5 ± 0.6	13.6 ± 0.6	−6.5 ± 0.5 ¶¥		
			CG (10)	14.6 ± 0.6	14.2 ± 0.6	−2.6 ± 0.5		0.843 (<0.0001)
		Girls	FG (10)	15.3 ± 0.9	14.4 ± 0.8	−6.1 ± 0.6 ¶¥		
			SG (10)	15.6 ± 1.0	14.7 ± 1.0	−6.2 ± 0.5 ¶¥		
			CG (10)	15.4 ± 0.7	15.0 ± 0.7	−2.4 ± 0.6		3.357 (<0.0001)
	U12	Boys	FG (10)	12.9 ± 0.5	11.7 ± 0.4	−9.3 ± 0.7 ¶†‡§		
			SG (10)	12.8 ± 0.8	11.6 ± 0.7	−9.2 ± 0.6 ¶†‡§		1.339 (<0.0001)
			CG (10)	12.9 ± 0.8	12.5 ± 0.8	−2.6 ± 0.5		
		Girls	FG (10)	14.5 ± 0.9	13.6 ± 0.4	−6.2 ± 0.7 ¶‡		0.843 (<0.0001)
			SG (10)	14.5 ± 0.6	13.6 ± 0.6	−6.0 ± 0.5 ¶‡		
			CG (10)	14.6 ± 0.6	14.2 ± 0.6	−2.6 ± 0.5		

Values are given as mean ± SD; U8: Under 8; U10: Under 10; U12: Under 12; FG: firm group; SG: sand group; CG: control group; Δ%: pre-post change percentage; ES: effect size; SHT: 4 × 10-m shuttle run performance. ¶ Significantly different from CG. † A significant difference when comparing boys to girls of FG and SG at U12, respectively. ‡ A significant difference when comparing U12 boys to U8 boys of FG and SG, respectively. § A significant difference when comparing U12 boys to U10 boys of FG and SG, respectively. ¥ A significant difference when comparing U10 boys/girls to U8 boys/girls of FG and SG, respectively. The statistical significance level was set at *p* ≤ 0.05.

## Data Availability

The data presented in this study are available on request from the corresponding author. The data are not publicly available due to privacy reasons.

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
