# Peer review of "Effects of Four Weeks of Plyometric Training Performed in Different Training Surfaces on Physical Performances in School Children: Age and Sex Comparisons"

_children, 2022, doi:10.3390/children9121914_

Round 1

Reviewer 1 Report

Dear Author, congratulations on your successful study. Overall, the manuscript is well written, and the results are very and convincing, but several minor corrections need to be made. My suggestions are presented in the appendix.

Author Response

Referee 1

Main comments:

Dear Author, congratulations on your successful study. Overall, the manuscript is well written, and the results are very and convincing, but several minor corrections need to be made. My suggestions are presented in the appendix. 

Author's response:

We thank the expert reviewer for his/her comments. Please find changes in the text.

Reviewer 2 Report

Dear authors, I enclose my comments:

Title

The title should be shortened by indicating method, sample and objective only. Recommendation.

General comments

- The term "sex" (biological) and not "gender" (cultural) should be used.

Abstract

The abstract adequately summarises the research, including the main findings of the results and the conclusions. However, it does not specify clearly and concretely the objective and methodology used.

Keywords:

Changing gender.

Introduction

The general idea of the research is presented with an adequate and up-to-date theoretical framework.

Materials and Methods

All procedures are correctly defined.

Reference authors should be included for each test. 

Results 

They are adequately presented.

Check the different tables. The data F, p & ES values, I would advise to place them by columns for a correct reading and visualisation (more agile).

This symbol is not defined or it is a misprint: ¶  

Discussion 

It is well contrasted and justified with the existing literature. 

Conclusions

They are clear.

Limitation

Reference should be made to the different types of sand (grammage, washed sand or not, wet or not, etc.). This is undoubtedly a limiting factor and can cause bias. It should therefore be reflected as an important aspect.

References

Journals should be in abbreviated format.

Check year and page format.

Check spacing between words. Double spaces or none at all.

Best regards.

Author Response

Referee 2

Comment 1

- The title should be shortened by indicating method, sample and objective only. Recommendation.

Author's response:

Thank you for your valuable comment. The title was changed as follows:

“Effects of four weeks of plyometric training performed in different training surfaces on physical performances in schoolchildren: Age and sex comparisons”

Comment 2

 - The term "sex" (biological) and not "gender" (cultural) should be used.

Author's response:

Thank you for your valuable comment.

The term “gender” was replaced by “sex”. Please find changes in the text.

Comment 3

- The abstract adequately summarises the research, including the main findings of the results and the conclusions. However, it does not specify clearly and concretely the objective and methodology used.

Author's response:

We thank the expert referee for his/her comment.

The objective and methodology were specified. Please find changes in the text.

Comment 4

- Keywords: Changing gender.

Author's response:

We thank the expert referee for his/her comment.

The term “gender” was replaced by “sex”. Please find changes in the text.

Comment 5

Results: They are adequately presented.

- Check the different tables. The data F, p & ES values, I would advise to place them by columns for a correct reading and visualisation (more agile).

Author's response:

We thank the expert referee for his/her comment.

F, p & ES values were placed in separated columns. Please find changes in the text.

- This symbol is not defined or it is a misprint: ¶  

Author's response:

Thank you for your valuable comment.

The symbol “¶” was defined.

Comment 6

- Limitation: Reference should be made to the different types of sand (grammage, washed sand or not, wet or not, etc.). This is undoubtedly a limiting factor and can cause bias. It should therefore be reflected as an important aspect.

Author's response:

We thank very much the expert referee for his/her comment.

“Third, reference should be made to the different types of sand (grammage, washed sand or not, wet or not, etc.). This is undoubtedly a limiting factor and can cause bias. It should therefore be reflected as an important aspect.”

Comment 7

Journals should be in abbreviated format.

Check year and page format.

Check spacing between words. Double spaces or none at all.

Author's response:

We thank the expert referee for his/her comment.

All references were checked and corrected according to journal guidelines.  Please find changes in the text.